# A Semantics-Based Approach for Simplifying IFC Building Models to Facilitate the Use of BIM Models in GIS

**Junxiang Zhu** [1] , **Peng Wu** [1,*] and **Chimay Anumba** [2]

1   School of Design and the Built Environment, Curtin University, Bentley, WA 6102, Australia;
    junxiang.zhu@curtin.edu.au
2   College of Design, Construction and Planning, University of Florida, Gainesville, FL 32611, USA;
    anumba@ufl.edu
*   Correspondence: peng.wu@curtin.edu.au; Tel.: +61-8-9266-4723

**Abstract:** Using solid building models, instead of the surface models in City Geography Markup Language (CityGML), can facilitate data integration between Building Information Modeling (BIM) and Geographic Information System (GIS). The use of solid models, however, introduces a problem of model simplification on the GIS side. The aim of this study is to solve this problem by developing a framework for generating simplified solid building models from BIM. In this framework, a set of Level of Details (LoDs) were first defined to suit solid building models—referred to as s-LoD, ranging from s-LoD1 to s-LoD4—and three unique problems in implementing s-LoDs were identified and solved by using a semantics-based approach, including identifying external objects for s-LoD2 and s-LoD3, distinguishing various slabs, and generating valid external walls for s-LoD2 and s-LoD3. The feasibility of the framework was validated by using BIM models, and the result shows that using semantics from BIM can make it easier to convert and simplify building models, which in turn makes BIM information more practical in GIS.

**Keywords:** Building Information Modeling (BIM); Geographic Information System (GIS); shapefile; Industry Foundation Classes (IFC); Level of Detail

## 1. Introduction

Geographic Information System (GIS) currently plays an important role in the development of smart cities, and in the center of smart cities are three-dimensional (3D) building/city models. The traditional source of building models for GIS is surveying [1], where airborne photogrammetry and laser scanning are used to capture city information, from which virtual city models can be created [2]. However, these data collection methods can only be applied to existing buildings [3], and it is costly and time-consuming to create city models in such a way, due to the need for manual processing of geometry and semantics of models [4–6].

An emerging alternative source of building models for the geospatial industry is the Building Information Modeling (BIM), which is a technology used in the Architecture, Engineering, and Construction (AEC) domain for creating, managing and sharing building information [1]. BIM models have rich geometric and semantic information about buildings and can be used in the whole life cycle of buildings, from the early planning and design, through construction and operation, to the end demolition [7]. Many countries or regions, such as the United Arab Emirates (UAE) and the United Kingdom (UK), have mandated the use of BIM for new buildings [8], which makes BIM an even more prominent source of building models for the geospatial industry.

BIM models need to be converted before they can be used in GIS, due to the use of different data standards in the AEC domain and the geospatial industry. Converting BIM models into surface-based CityGML (City Geography Markup Language) models is frequently investigated. Various methods have been developed to generate CityGML

models from BIM models [6,9,10]. Another concern during data conversion is that the converted building models can be over-detailed for applications in the geospatial industry, because the original BIM models contain rich geometric and semantic information [11,12]. Over-detailed building models consume more storage space and more computing resources for visualization and data transmission [13]. Therefore, BIM models need to be simplified, and the main approach for surface-based CityGML models is to apply the concept of Level of Detail (LoD) [14]. According to Mignard et al. [15], Zhou et al. [13] and Deng et al. [16], LoD can help decrease the complexity of object geometries, improve the efficiency of visualization, and reduce data storage.

An alternative way to use building information in GIS is to convert BIM models into solid building models, instead of surface models, due to easier geometry conversion and more flexible semantics transfer [17]. The adoption of solid models in GIS, however, introduces the problem of building model simplification. Due to the difference in modeling paradigm [18], simplification methods developed for surface models cannot be directly applied to solid models, and there is currently little research into simplifying solid building models. Therefore, the aim of this study is to develop a framework to simplify BIM models and generate solid building models on the GIS side. The developed framework is expected to bring in a new paradigm in representing BIM information in GIS, which would make BIM information easier and more practical to use.

## 2. Related Work

### 2.1. Building Model Simplification in the Geospatial Industry

Building model simplification, or generalization, has been studied by the geospatial industry in the past decades, mainly for the purpose of multi-scale representation. Before the advent of CityGML, studies on model simplification were mainly focused on the simplification of geometry by removing redundant geometric parts such as extrusion and intrusion, where points and edges might be removed to generalize building geometry. For example, Kada [19] used the normal of planes to detect and remove wall extrusion. Forberg [20] simplified orthogonal buildings by removing parallel facets. After the advent of CityGML, semantics was also considered during model generalization, but the focus of simplification was still on the geometry. For example, Fan et al. [21] developed methods for extracting the exterior shell of a building to simplify ground plan and building façade. Baig et al. [22] developed a three-step strategy for the generalization of building ground plans by removing unnecessary edges. Li and Nan [23] developed a general method to simplify mesh building models that are from point clouds and aerial images by detecting and removing excessive points. Another characteristic of model simplification in the geospatial industry is that models with low LoDs were derived from existing high-LoD models. For example, in the study by Baig et al. [22], LoD1 models were derived from LoD3 models. Kim and Li defined LoD for indoor space as GLoD-I (geometric LoD-Interior) and converted indoor space models into various GLoD-Is from GLoD-I0 to GLoD-I3 [24].

In a nutshell, in the geospatial industry, building model simplification mainly refers to converting LoD and simplifying the geometry of certain building components, in which high-LoD models are converted into low-LoD models and unnecessary points and/or edges are removed to generalize the shape of building. In this process, both geometric information and semantic information are involved and simplified.

### 2.2. Building Model Simplification in the Context of BIM/GIS Integration

#### 2.2.1. BIM/GIS Integration

In recent years, BIM/GIS integration is attracting attention from researchers in the AEC domain and the geospatial industry [25]. In general, studies on BIM/GIS integration fall into two categories, i.e., data integration (data-level integration) and application (application-level integration). Data-level integration is the foundation of application-level integration. For data-level integration, while there are many ways to integrate BIM and GIS data, the main stream is to use building information in a GIS environment [26]. This

requires BIM models to be converted into a GIS-compatible standard/format. Industry Foundation Classes (IFC) is the representative for BIM, which is an open international standard for data exchange in the AEC domain [7], whereas CityGML and shapefile are representatives for GIS. CityGML is an open data model released by OGC (Open Geospatial Consortium) for the storage and exchange of virtual 3D city models [27], while shapefile is a spatial data format developed by ESRI (Environmental Systems Research Institute) for the storage and exchange of general spatial features [28], which is able to store both 2D and 3D data.

Accordingly, the two most common paths for data conversion are the IFC-to-CityGML path and the IFC-to-shapefile path. Both of these paths need to deal with two major tasks, including geometry conversion and semantics transfer. Geometry conversion deals with the conversion of all types of IFC representations into a form that is accessible by GIS, whereas semantics transfer deals with the transfer of semantic information from BIM to GIS. Please note that the use of different data formats is essentially the use of different model types in GIS, i.e., surface models versus solid models.

Based on the achievements at data integration, BIM and GIS have been jointly applied in various applications, such as room-level traffic noise assessment [9] and flood damage assessment [29], bridge management [30], offshore platforms disassembly [31], green building [32], building retrofit [33], construction site layout optimization [34] and construction supply chain management [35].

### 2.2.2. Surface Building Model Simplification in BIM/GIS Integration (IFC-to-CityGML)

In the context of BIM/GIS integration, when BIM models are converted into surface models, the simplification of models is mainly realized by converting IFC models into CityGML models at various LoDs.

IFC models contain rich building information, and the amount of information in BIM models can be indicated by the level of development (LOD) [36]. These two terms, LoD and LOD, have different subjects of interest. LOD focuses more on individual building elements [37], whereas LoD focuses more on the holistic building. For IFC-to-CityGML conversion, LoD conversion is an indispensable part of data conversion.

LoD conversion contains two tasks, including semantics mapping and geometry conversion. The main tasks for semantics mapping and geometry conversion are presented in Figure 1. The first part addresses the question of what classes (or building elements) should be retained in a specific LoD, as IFC classes and CityGML classes do not have a clear one-to-one mapping, while the second deals with the solid-to-surface conversion. These two problems have been investigated by many studies [38]. For example, Isikdag et al. [39] developed a conceptual framework for the generation of CityGML building models from IFC datasets, where detailed steps regarding class mapping, geometry simplification, and semantic information transfer for each LoD have been discussed. Donkers et al. [6] developed a sophisticated algorithm for automatic conversion of IFC datasets into geometrically correct CityGML LoD3 buildings. Deng et al. [16] developed the Semantic City Model, which has taken LoD conversion into consideration when converting IFC into CityGML. Kang and Hong [40] used the Screen-Buffer scanning-based Multiprocesing (SB-MP) method to automatically generate LoD1 to LoD4 CityGML models.

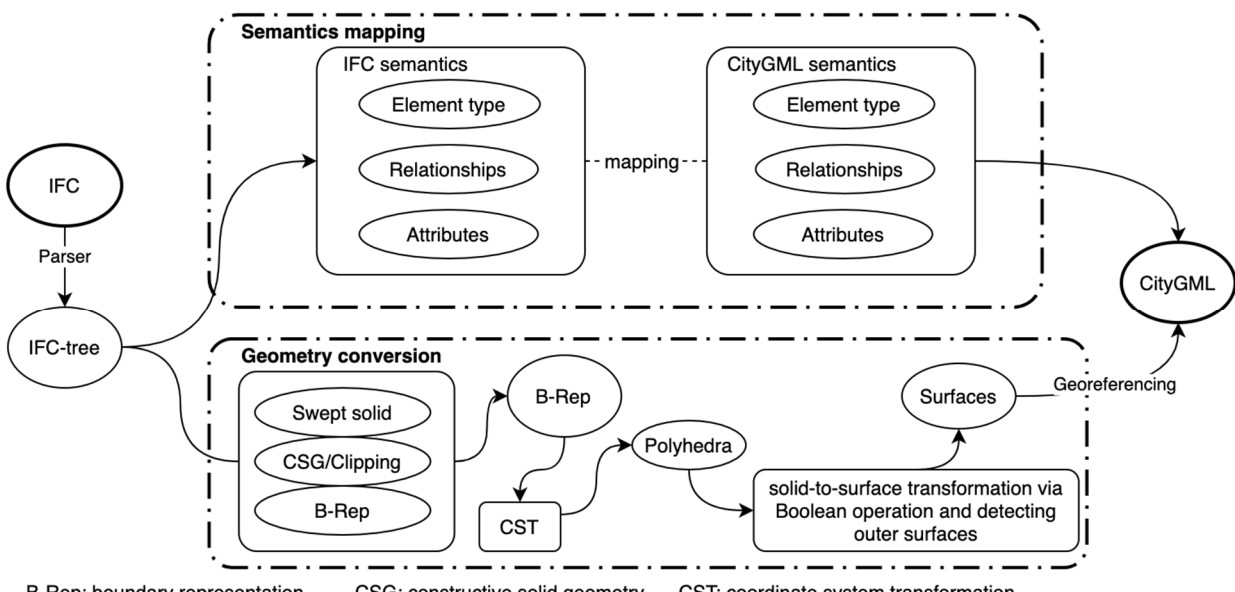

**Figure 1.** Tasks in conversion/simplification from Industry Foundation Classes (IFC) to City Geography Markup Language (CityGML).

### 2.2.3. Solid Building Model Simplification in BIM/GIS Integration

Compared with surface models, there is little research into the simplification of solid building models. This can be attributed to the fact that surface models are more prevalent than solid models in the geospatial industry, because (1) GIS has a limited capability in 3D modeling [41], it is not efficient in creating solid 3D models, especially complex ones, and (2) the use of airborne photogrammetry and laser scanning [3] in the geospatial industry determines that surface models, especially those of low LoD, can be more effectively created.

To make it easier to use BIM information in GIS, some researchers investigated the possibility of using solid models. The main efforts were from Amirebrahimi et al. [29,42], Zhu et al. [17,18,43,44], Boyes et al. [45], and Isikdag et al. [46]. These studies realized the conversion of IFC into shapefile-based or Geography Markup Language (GML)-based solid models, using either self-developed tools/algorithms or commercial tools such as the Data Interoperability Extension for ArcGIS (DIA) (Redlands, CA, USA) [47] and the Feature Manipulation Engine (FME) (Surrey, BC, Canada) [48]. However, these studies did not take model simplification into consideration [17].

## 3. Materials and Methods

### 3.1. Semantics-Based Approach for Solid Model Simplification

The literature review has suggested that there was a lack of study on simplifying solid building models in the context of BIM/GIS integration. This study proposed a semantics-based approach for solid model simplification. Semantic information, or information other than geometry information, is an important part of building information, which is one of the features that distinguishes BIM models from geometry-centered CAD (computer aided design) models [7]. As suggested by Donkers et al. [6], semantic information can be useful during model conversion/simplification.

#### 3.1.1. Semantic Information in IFC

A thorough investigation into the IFC standard revealed that there are two hierarchical structures in IFC, i.e., the class hierarchy for managing IFC classes and the spatial structure [49] which shows topology information. These two structures determine the semantic information in IFC.

The class hierarchy manages all the IFC classes in a hierarchical structure. Everything in IFC is treated as an object under a specific class. Every class has a parent (or superclass), except the root class, and every class may have one or more child (or subclass), except the leaf classes. A class contains multiple attributes, and child classes tend to have more attributes than their parent, as they can define their own attributes in addition to those inherited from their parent.

The spatial structure defines the topology of IFC building models using four spatial structure elements (or spatial containers) including site, building, building story and space. These spatial structure elements are containers for building elements. Figure 2 presents a typical spatial structure of IFC data. Through this spatial structure, spatial structure elements and building elements are linked to form a huge network.

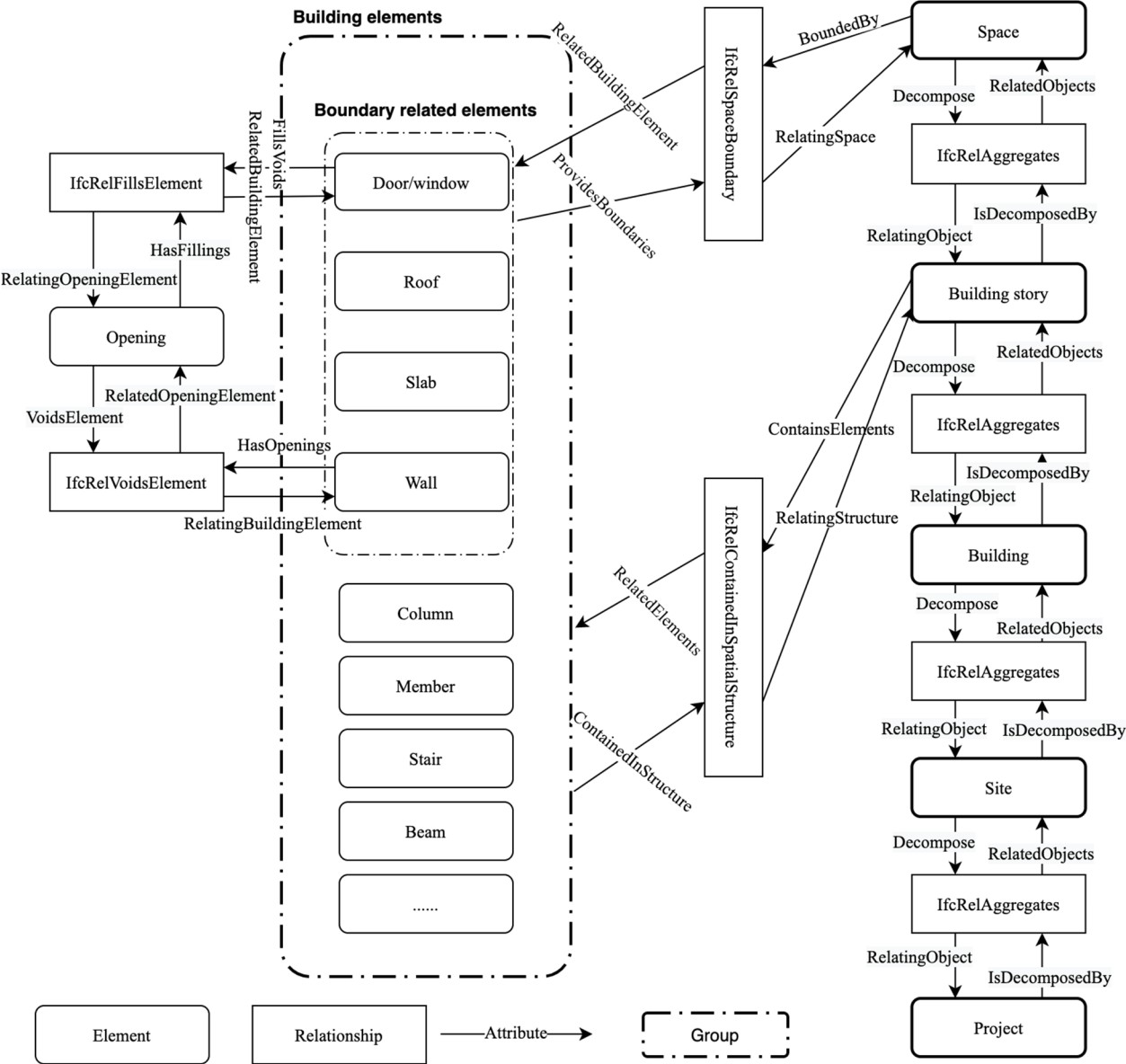

**Figure 2.** A typical spatial structure of IFC data.

### 3.1.2. Geometric Information in IFC

In IFC, 3D representations of building elements and spatial structure elements are mainly presented using sweeping, Constructive Solid Geometry (CSG)/Clipping, and boundary representation (B-Rep) [43]. Clipping is a subtype of CSG where only the difference operation is used. By a close examination of each building element and spatial structure element in IFC [49], the modeling methods for each building elements are summarized in Table 1. Sweeping and CSG/Clipping are implicit representations, where a set of geometric parameters are used to implicitly represent building elements, and the final explicit shapes (B-Rep) of building elements have to be generated from these geometric parameters before being displayed.

**Table 1.** Building elements, spatial elements, and supported representations.

| IFC Classes | | Representation Type/Modeling Method | | | |
| --- | --- | --- | --- | --- | --- |
| | | Sweeping | CSG/Clipping | B-Rep | Not Specified |
| Building elements | IfcBeam, IfcColumn, IfcMember, IfcPlate, IfcRamp, IfcRampFlight, IfcSlab, IfcWall | x | x | | |
| | IfcCovering, IfcStair, IfcStairFlight | x | | | |
| | IfcDoor | x | | x | |
| | IfcWindow, IfcCurtainWall | | | x | |
| | IfcBuildingElementProxy, IfcChimney, IfcFooting, IfcPile, IfcRailing, IfcRoof, IfcShadingDevice | | | | x |
| Spatial structure elements | IfcSite, IfcBuilding, IfcBuildingStorey | | | x | |
| | IfcSpace | x | x | x | |

### 3.1.3. The Overall Framework of This Study

This study developed a framework for converting and simplifying solid building models at two levels, i.e., building level and component level, as shown in Figure 3. (1) At building level, a set of LoDs were specifically defined for solid building models, indicating the building elements that should be included in each LoD. In order to distinguish it from the LoD in CityGML, the LoDs defined in this study for solid models are referred to as s-LoD. The building-level model simplification is mainly about building component filtering. (2) At building component-level, the geometry of building elements, such as doors and windows, was simplified. Component-level model simplification is mainly about geometry simplification. Finally, via IFC-to-shapefile conversion, the conceptually simplified building models can be generated and stored in shapefile.

### *3.2. Building-Level Model Simplification*

#### 3.2.1. Defining s-LoD and Grouping Building Elements for Solid Models

When defining s-LoD, this study referenced the LoD concept defined by CityGML, which has five levels of detail from LoD0 to LoD4 [27]. The s-LoD0 models are footprint or roof edge polygons. The s-LoD1 models are block models with flat roof structures. Models at s-LoD2 have differentiated roof structures and thematically differentiated boundary building elements. The s-LoD3 models are architectural models with detailed wall and roof structures potentially including doors and windows. An s-LoD4 model is based on an s-LoD3 model, but with interior structures. Based on this conceptual framework of s-LoD, building elements for each s-LoD are listed in Table 2. s-LoD is different from LoD. LoDs are based on surface models, while s-LoDs are applied to solid models. For example, a solid external wall in s-LoD includes an external surface wall and an internal surface wall in LoD.

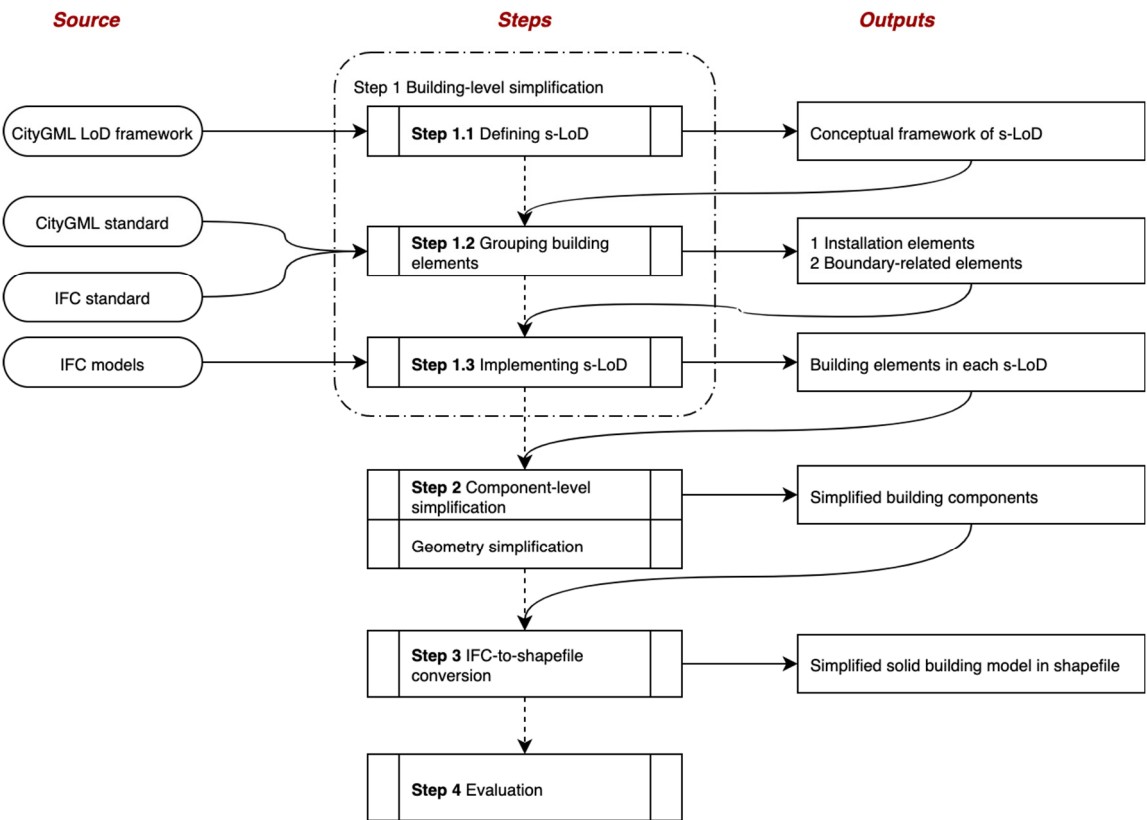

**Figure 3.** The overall framework for solid building simplification.

**Table 2.** The s-LoD (Level of Detail) definitions for solid building models.

| Level | Sketch | Contained Building Elements |
|-------|--------|-----------------------------|
| s-LoD0 |  | Footprint or roof edge. |
| s-LoD1 |  | Footprint (of base slab) extruded to the height of building. |
| s-LoD2 |  | Roof, base slab, and external wall. |
| s-LoD3 |  | Roof, base slab, external wall, external door, external window, and other external objects. |
| s-LoD4 |  | All elements in the original BIM model. |

A semantic data model has been developed for solid building models based on this s-LoD concept and is presented in Figure 4 in the form of Unified Modeling Language (UML) diagram. This semantic data model has features from both the IFC data model and the CityGML data model. It inherits the spatial structure from IFC, and groups building elements into two broad classes, i.e., the boundary solids (_BoundarySolid) for boundary-related building elements and the installation solids (_InstallationSolid) for other building elements. As with CityGML, this data model put more weight on the boundary-related building elements, including slab, wall, roof, window, and door.

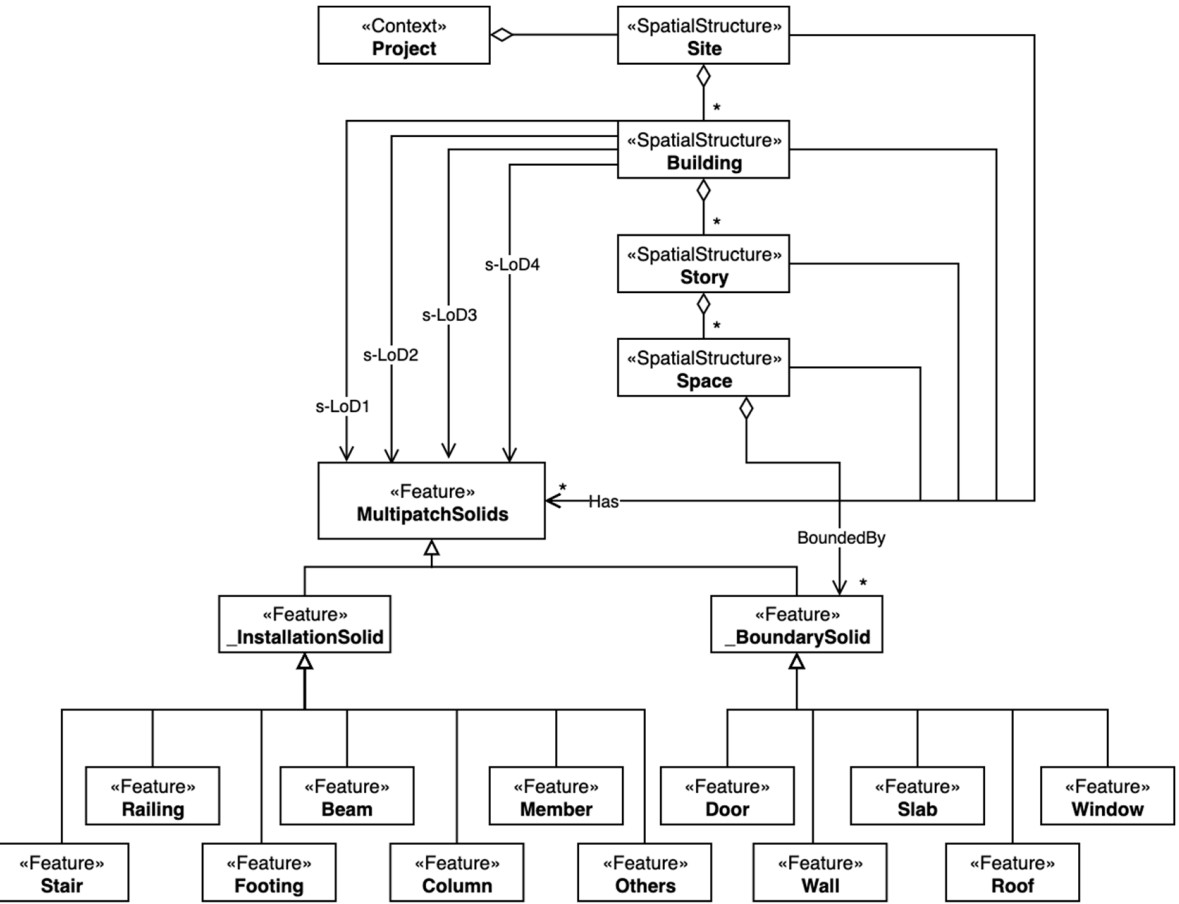

**Figure 4.** The Unified Modeling Language (UML) diagram of the semantic data model for the solid building model.

3.2.2. Implementing s-LoDs

Based on this semantic data model, an overall framework for generating s-LoD1 to s-LoD4 models was developed, as shown in Figure 5. For s-LoD4, all building components can be individually generated. Models of s-LoD1 can be generated using information from the base slab and the building height. Models of s-LoD2 were produced using raw geometric information and semantic information from roofs, base slabs and external walls. Lastly, s-LoD3 models were created using information from external doors, external windows, roof, base slabs and external walls.

While challenges in surface model simplification come from exterior shell extraction and opening removal for generating LoD2 and LoD3 models [16], the unique challenges in solid model simplification and generation mainly come from the following aspects, including (1) identifying external objects for s-LoD2 and s-LoD3; (2) distinguishing various slabs, which are used for different purposes in construction, such as roofs, floors and base slabs; and (3) generating valid external walls for s-LoD2 and s-LoD3.

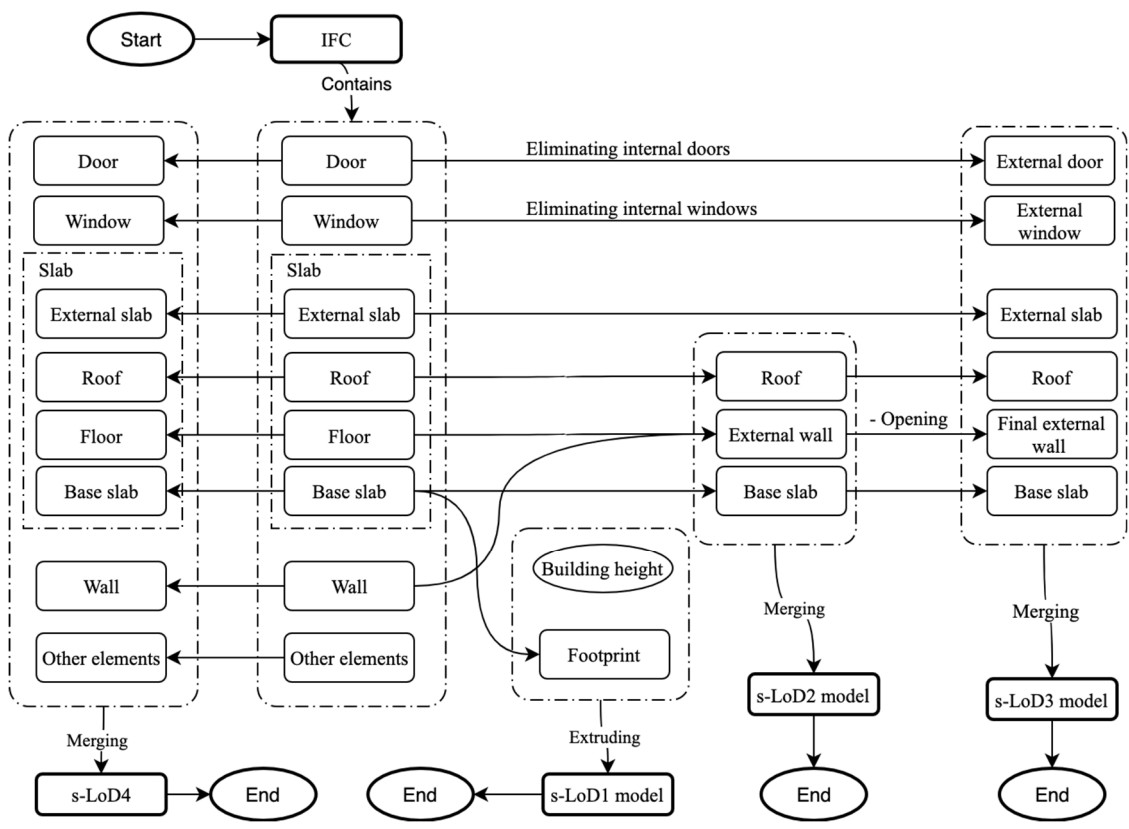

**Figure 5.** Workflow for generating s-LoD1 to s-LoD4 models.

(1)    Identifying external objects

For a specific building element, external objects should be identified for s-LoD2 and s-LoD3, and a reliable method for distinguishing external objects is essential to implementing s-LoDs. After a thorough investigation into the IFC standard, it can be noted that two methods can be used for this purpose, i.e., using the *IsExternal* attribute of building elements and using the *IfcRelSpaceBoundary* relationship. The first method is straightforward and covers most of the building elements, as shown in Table 3. Among the 21 building elements defined in IFC4, 17 building elements have defined the *IsExternal* attribute.

**Table 3.** Industry Foundation Classes (IFC) building elements with the "IsExternal" attribute.

| IFC Class | "IsExternal" Defined? | IFC Class | "IsExternal" Defined? | IFC Class | "IsExternal" Defined? |
|---|---|---|---|---|---|
| IfcBeam | Yes | IfcFooting | No | IfcRoof | Yes |
| IfcBuildingElementProxy | Yes | IfcMember | Yes | IfcShadingDevice | Yes |
| IfcChimney | Yes | IfcPile | No | IfcSlab | Yes |
| IfcColumn | Yes | IfcPlate | Yes | IfcStair | Yes |
| IfcCovering | Yes | IfcRailing | Yes | IfcStairFlight | No |
| IfcCurtainWall | Yes | IfcRamp | Yes | IfcWall | Yes |
| IfcDoor | Yes | IfcRampFlight | No | IfcWindow | Yes |

The second method is to use the *IfcRelSpaceBoundary* relationship, which is a relationship between building elements and space (*IfcSpace*). The relationship between building elements, *IfcRelSpaceBoundary*, and *IfcSpace* is presented in Figure 6. The *IfcRelSpaceBoundary* relationship has an attribute, *InternalOrExternalBoundary*, indicating whether a boundary is external or not. This method, however, can only be applied to boundary-related building elements, such as doors, windows, walls, roofs, and slabs.

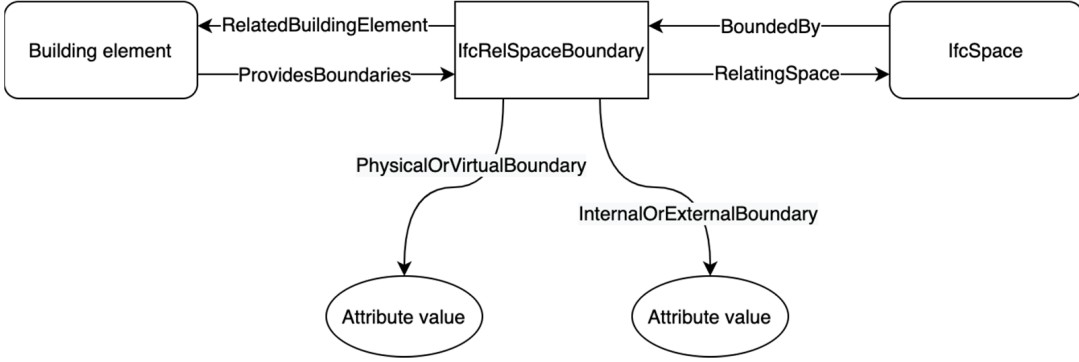

**Figure 6.** Relationship between building element, *IfcRelSpaceBoundary* and *IfcSpace*.

(2)  Distinguishing between slabs

A method for distinguishing various slabs is necessary, as slabs can be used internally or externally for various purposes in buildings, such as floors (internal) and roofs (external) [49]. After a close investigation into the *IfcSlab* class, a decision-tree-based method was developed to differentiate slabs, which is presented in Figure 7. This method utilizes several types of semantic information, including the class of an object, its predefined type, whether the object is a boundary, and the building story in which the object resides. Using this method, one is able to distinguish between roofs, floors, external slabs and base slabs. The detailed criteria used for distinguishing slabs are listed in Table 4.

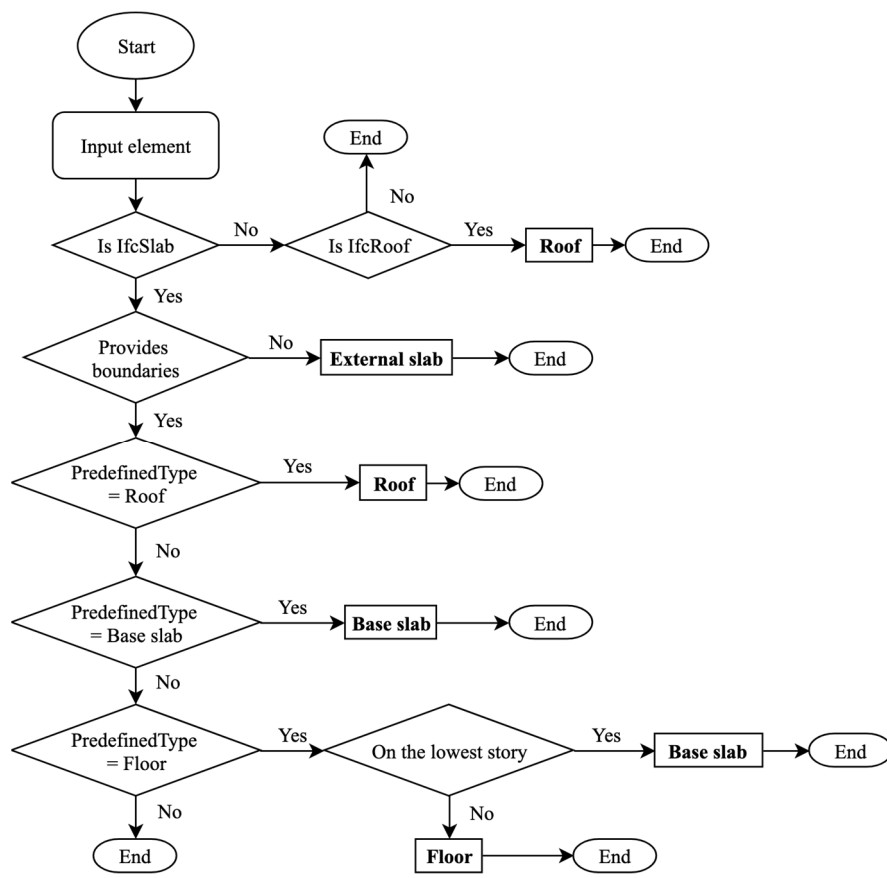

**Figure 7.** Decision tree for determining slab types, i.e., base slabs, floors, external slabs and roofs.

**Table 4.** Criteria for differentiating roofs, floors, external slabs, and base slabs.

| Types | s-LoD | Criteria |
|---|---|---|
| Base slab | s-LoD2, s-LoD3, s-LoD4 | 1. IfcSlab, providing boundary to a space and the predefined type is base slab, or 2. IfcSlab, the predefined type is floor and on the lowest building story. |
| External slab | s-LoD3 | 1. IfcSlab, not providing boundaries to space. |
| Floor | s-LoD4 | 1. IfcSlab, providing boundary to a space, the predefined type is floor and not on the lowest building story. |
| Roof | s-LoD2, s-LoD3, s-LoD4 | 1. IfcRoof, or 2. IfcSlab, providing boundary to a space and the predefined type is roof. |

(3)    Generating valid external walls for s-LoD2 and s-LoD3

External walls for s-LoD2 and s-LoD3 models need more attention than those for s-LoD4, because of the void that results from from the removal of internal floors, doors, and windows from s-LoD2 and/or s-LoD3 models. An example is given in Figure 8. If the traditional high-to-low simplification approach is to be applied, complex geometric algorithms would be needed to fill in these openings to derive valid s-LoD3 and s-LoD2 walls as shown in Figure 8b,c. In this study, this problem was solved in an easier manner by sweeping the external walls by an extra distance. The total sweeping depth for the external wall is the sum of the original sweeping depth ($H_{wall}$) and the thickness of the floor above ($H_{floor}$).

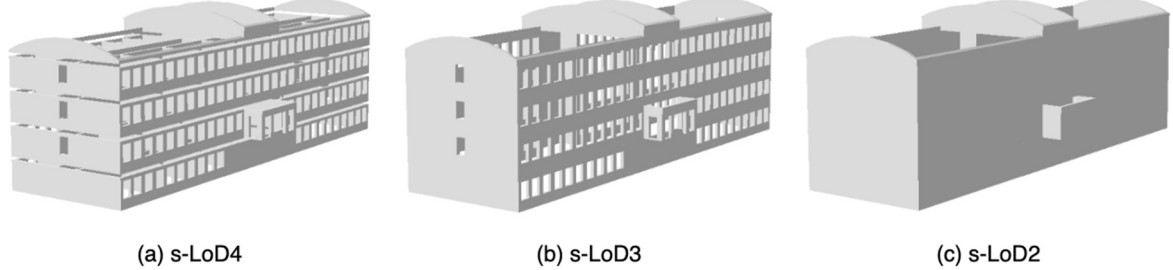

(a) s-LoD4                              (b) s-LoD3                              (c) s-LoD2

**Figure 8.** External walls for s-LoD4, s-LoD3, and s-LoD2 models. (**a**) s-LoD4. (**b**) s-LoD3. (**c**) s-LoD2.

$H_{floor_i}$ and $H_{wall_i}$ for the $i$th story can be obtained by using the workflow presented in Figure 9. (1) For obtaining $H_{wall_i}$, the bounding box of each wall in the $i$th story is retrieved. Bounding boxes are another type of geometric representation in IFC for building objects, which are defined by three values, namely $x\_dim$, $y\_dim$ and $z\_dim$ indicating the length in $x$-axis, $y$-axis and $z$-axis, respectively. The $H_{wall_i}$ is then the maximum $z\_dim$ of all the wall bounding boxes in the $i$th story. (2) For obtaining $H_{floor_i}$, as floors (slabs) are represented using swept solid, the $H_{floor_i}$ is then the sweeping depth of the floor. The building story height, $H_{story_i}$, can then be calculated using the equation as follows:

$$H_{story_i} = H_{floor_i} + H_{wall_i} \tag{1}$$

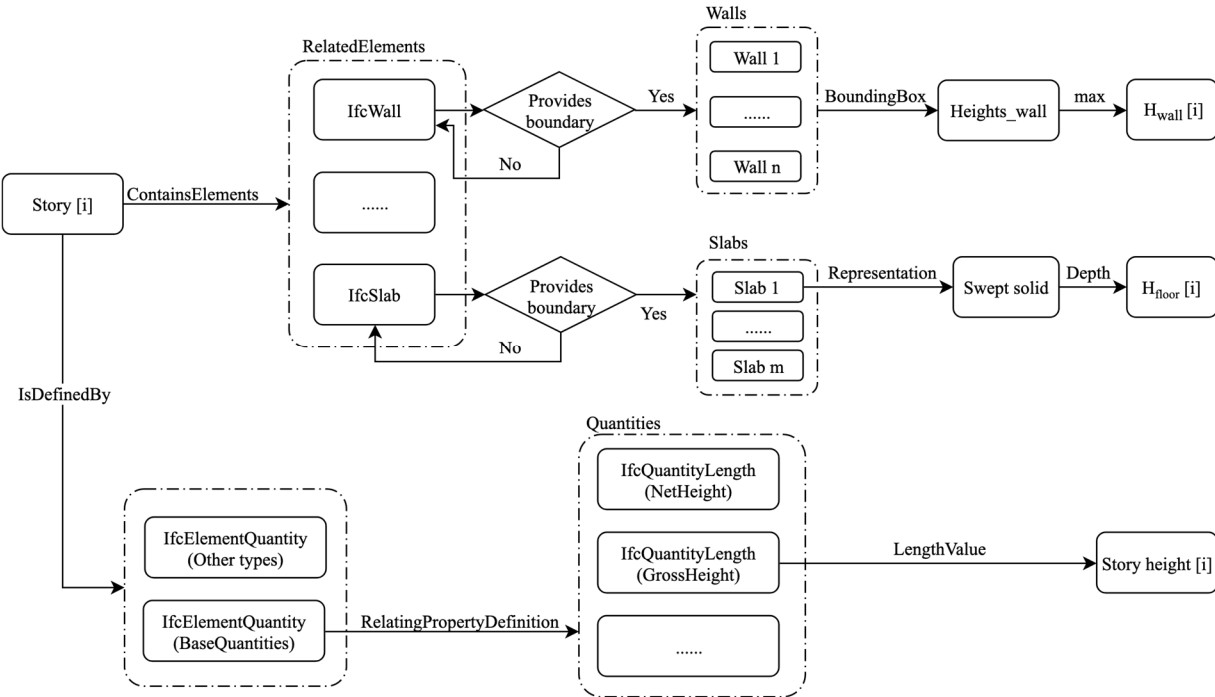

**Figure 9.** Acquiring building story height using two methods.

In IFC, another attribute of the building story class, *GrossHeight*, is used to indicate building story height. For a semantically correct BIM model, the building story height from these two sources should match.

### 3.3. Component-Level Simplification

### 3.3.1. Simplification of Doors and Windows

Doors and windows are simple objects, but their geometry can be over-detailed; in some cases, a door object may contain as many as 32 components [43], especially when B-Reps are used.

In IFC, doors and windows are connected to walls (not curtain walls) via opening elements; doors and windows thus share the same profile with corresponding openings. Please note that, as with building elements, openings in IFC are also objects with individual geometric representations, while the opening in CityGML is only an abstract class. Based on this fact, a novel method was developed to simplify B-Rep doors and windows, where the profiles of openings were used.

Figure 10 shows the related IFC classes and their relationships for this method. The required parameters for generating simplified doors and windows were obtained from two classes, i.e., openings and bounding boxes of windows/doors. Doors and windows are linked to openings via the *IfcRelFillsElement* relationship. The sweeping profile and sweeping direction can be retrieved from openings, while the sweeping depth can be obtained from the bounding box by using the following equation:

$$Depth = \min(x\_dim,\ y\_dim,\ z\_dim). \tag{2}$$

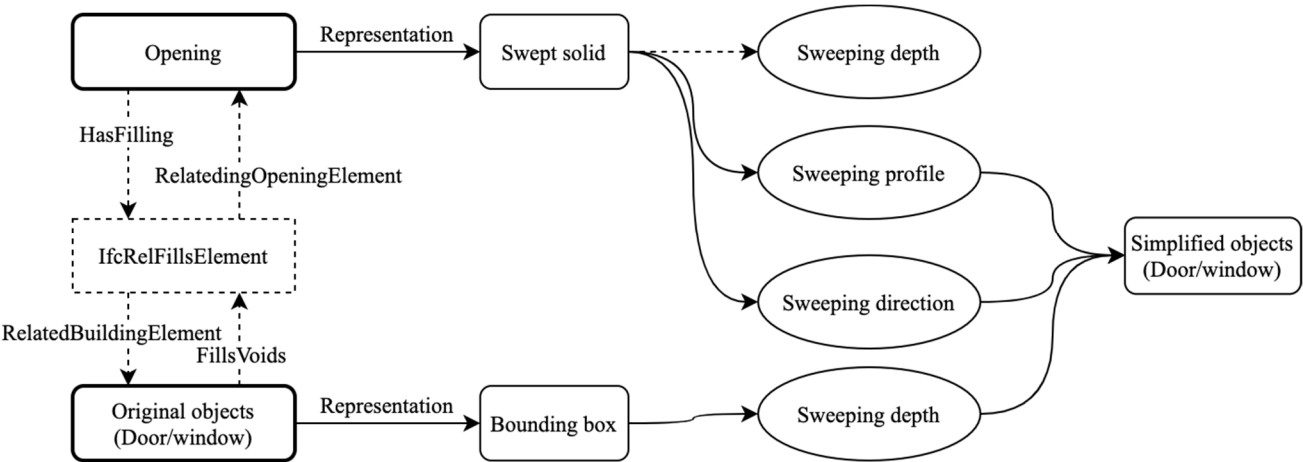

**Figure 10.** IFC classes involved in the simplification of doors and windows.

After all the necessary parameters are in place, the simplified windows and doors can be regenerated by sweeping.

### 3.3.2. Simplifying External Walls for s-LoD2 and s-LoD3

If a building has identical floor plans for each floor, such as the Smiley building, its external walls can be further simplified. To achieve this, when the traditional high-to-low simplification approach is adopted, the complex dilation and erosion algorithms would be used to merge such walls. In this study, this process was completed in an easier manner with the help of semantic information. The sweeping profile of external walls on the lowest story, as well as story heights and floor thickness, can be jointly used to generate simplified external walls by sweeping. The sweeping depth can be calculated by using the following equation:

$$Depth = \sum H_{floor_i} + \sum H_{wall_i}, \tag{3}$$

where $H_{floor_i}$ is the thickness of the floor on the *i*th story, and $H_{wall_i}$ is the height of walls on the same story.

### 3.4. IFC-to-Shapefile Conversion

Shapefile is adopted in this study as medium for solid building models for two reasons. (1) Shapefile is the native format of ArcGIS [50], which is the most frequently used GIS platform when integrating BIM information [26], and (2) shapefile has proven its effectiveness and efficiency in accommodating BIM models in many studies [42,45,51,52].

In the IFC-to-shapefile conversion, there are two main tasks, including geometry conversion and semantics transfer, as shown in Figure 11. Each task has several subtasks. For transferring semantics, the method proposed by Zhu et al. [17] was used. For the geometry conversion, there are three compulsory subtasks, including representation conversion, coordinate system transformation (CST) and geo-referencing.

Among those subtasks for geometry conversion, representation conversion is the most essential one. In spite of the fact that surfaces are allowed, IFC mainly uses solids to represent building components. Solids in IFC can be implicitly represented by parametric modeling methods, including sweeping, CSG/Clipping, or explicitly represented by B-Rep. These implicit parametric representations are commonly used in IFC [53] and have to be converted into explicit B-Rep [6] before they can be further processed. The conversion of sweeping and clipping to B-Rep has been carried out by [18,43,44,54], and the processes are briefly presented in Figure 12a,b, respectively. To convert swept solids, the sweeping profile (which consists of either a set of parameters for defining a shape or explicit points of a shape) and sweeping path (including the direction of sweeping and sweeping depth) have to be acquired from IFC datasets first, and then the methods developed by [44] can be

used to generate the corresponding B-Rep. To convert the clippings, which are Boolean difference between a swept solid and a half space solid, parameters for the swept solid and the half space should first be obtained from the IFC datasets in order to calculate their shapes, and then the Boolean difference should be applied to generate the final B-Rep, as presented in [18].

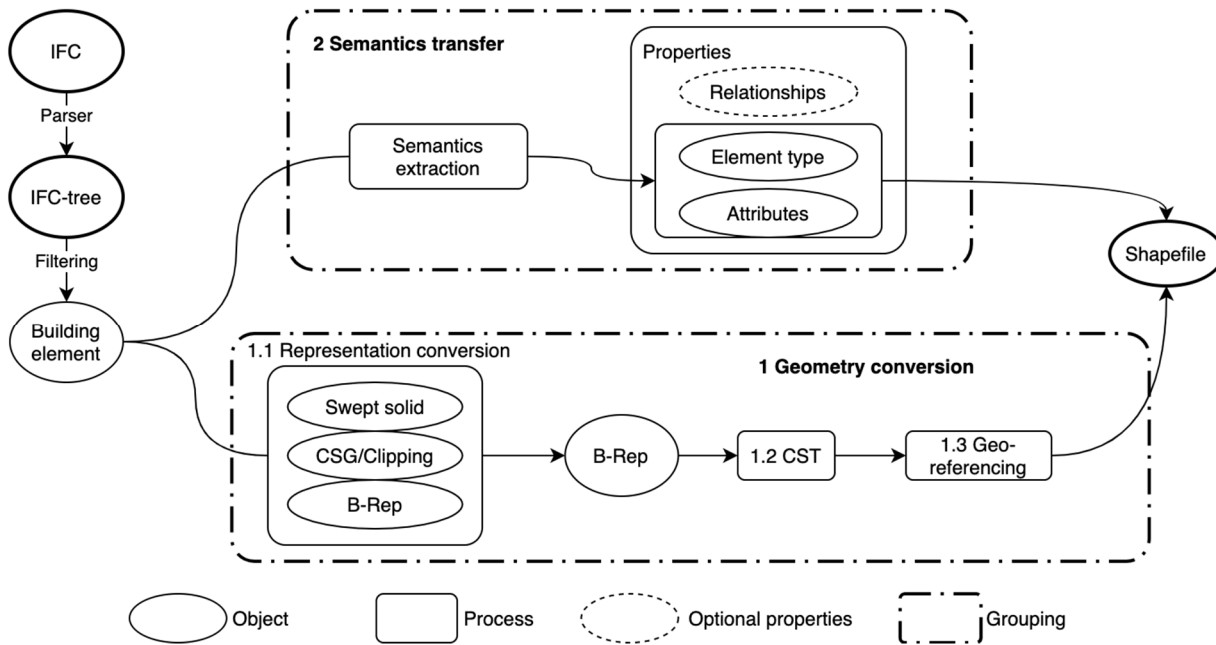

**Figure 11.** IFC-to-shapefile conversion.

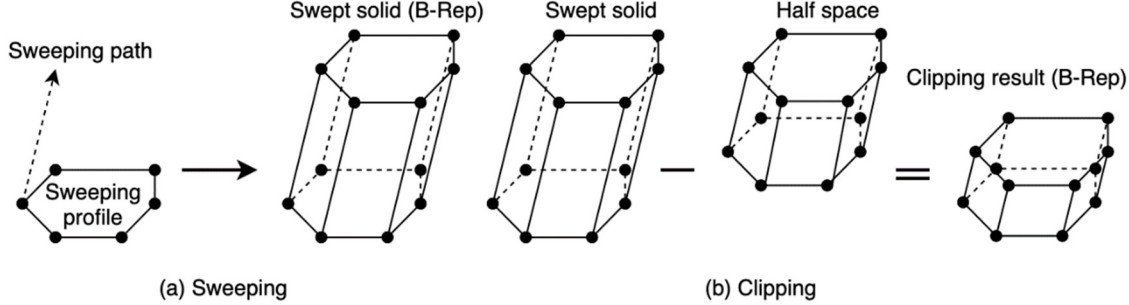

**Figure 12.** Converting Sweeping and Clipping to B-Rep. (**a**) Sweeping. (**b**) Clipping.

### 3.5. Method Evaluation

Three IFC building models, representing different building types, were used to assess the proposed framework, including a house model, an institute building model and the model of an apartment building named Smiley, as shown in Figure 13. These models are available online and can be used unrestrictedly [55]. They were selected in this study because (1) these models contain the most common building elements (see Table 5), especially boundary-related ones, representing varying building types and model sizes, and (2) they were built as (relatively) good examples of quality IFC models, in terms of both geometry and semantics. Please note that BIM models from actual projects were not used for validation in this study, because these models contain project-specific semantics, which may not suit the need for method validation. Not to mention that not all owners would share their models due to confidential concerns.

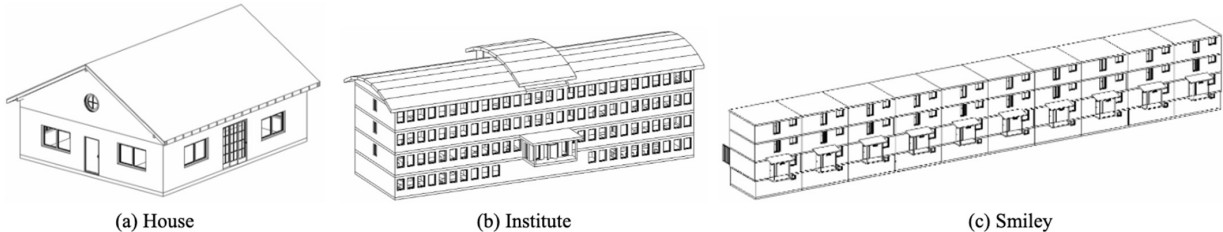

**Figure 13.** Models used, including (**a**) house model, (**b**) institute model and (**c**) Smiley building model.

**Table 5.** Building elements and component quantity of the house model, institute model and Smiley building model.

| IFC Class | Quantity of Components | | |
| --- | --- | --- | --- |
| | House | Institute | Smiley |
| IfcBeam | 4 | - | 10 |
| IfcColumn | - | 2 | 20 |
| IfcDoor | 5 | 77 | 170 |
| IfcMember | 42 | - | - |
| IfcRailing | 2 | 12 | 120 |
| IfcRoof | - | - | - |
| IfcSlab | 4 | 26 | 120 |
| IfcStair | 1 | 4 | 30 |
| IfcWall | - | - | 281 |
| IfcWallStandardCase | 13 | 121 | 270 |
| IfcWindow | 11 | 206 | 80 |
| Total | 82 | 448 | 831 |

The effectiveness of the simplification approach was assessed by file size, at both building and component levels, which is the most common criteria used by many similar studies [6]. For the whole building, two types of relatively reduced size (RRS) were used to assess the effectiveness of the methods, including *RRS*1 and *RRS*2. For building components, the file sizes before and after simplification were compared.

*RRS*1 is calculated by comparing a given s-LoD with s-LoD4 using

$$RRS1_i = (S_{s-LoD_4} - S_{s-LoD_i})/S_{s-LoD_4} \times 100\%, \tag{4}$$

where $RRS1_i$ is the *RRS*1 of the *i*th s-LoD, $S_{s-LoD_4}$ is the absolute file size of s-LoD4 model, and $S_{s-LoD_i}$ is the absolute file size of the *i*th s-LoD. Additionally, *RRS*2 is calculated by comparing adjacent s-LoDs, using

$$RRS2_i = (S_{s-LoD_{i+1}} - S_{s-LoD_i})/S_{s-LoD_{i+1}} \times 100\%, \tag{5}$$

where $RRS2_i$ is the *RRS*2 of the *i*th s-LoD. This reflects the relative amount of file size that has been reduced from the higher s-LoD.

## 4. Results

### 4.1. Generated Solid Building Models

Using the proposed framework, the s-LoD1 to s-LoD4 models for these buildings were generated, as shown in Table 6, where both internal view and external view are provided. A portion of external walls, windows and doors have been removed to show the interior. These models have been converted further into scene layer packages (SLPK) using ArcGIS Pro and uploaded to ArcGIS Online. SLPK is the format for the Indexed 3d Scene layer (I3S), which is an OGC Community Standard for streaming large 3d datasets. These models can be interactively viewed via the link (https://arcg.is/01SyDe, accessed on 27 October 2021), and a screenshot is provided in Figure 14. Please note that only the Smiley models have the correct location information.

**Table 6.** Generated models from s-LoD1 to s-LoD4 for house, institute and Smiley building.

| | House | | Institute | | Smiley | |
|---|---|---|---|---|---|---|
| | **External** | **Internal** | **External** | **Internal** | **External** | **Internal** |
| s-LoD1 | | | | | | |
| s-LoD2 | | | | | | |
| s-LoD3 | | | | | | |
| s-LoD4 | | | | | | |

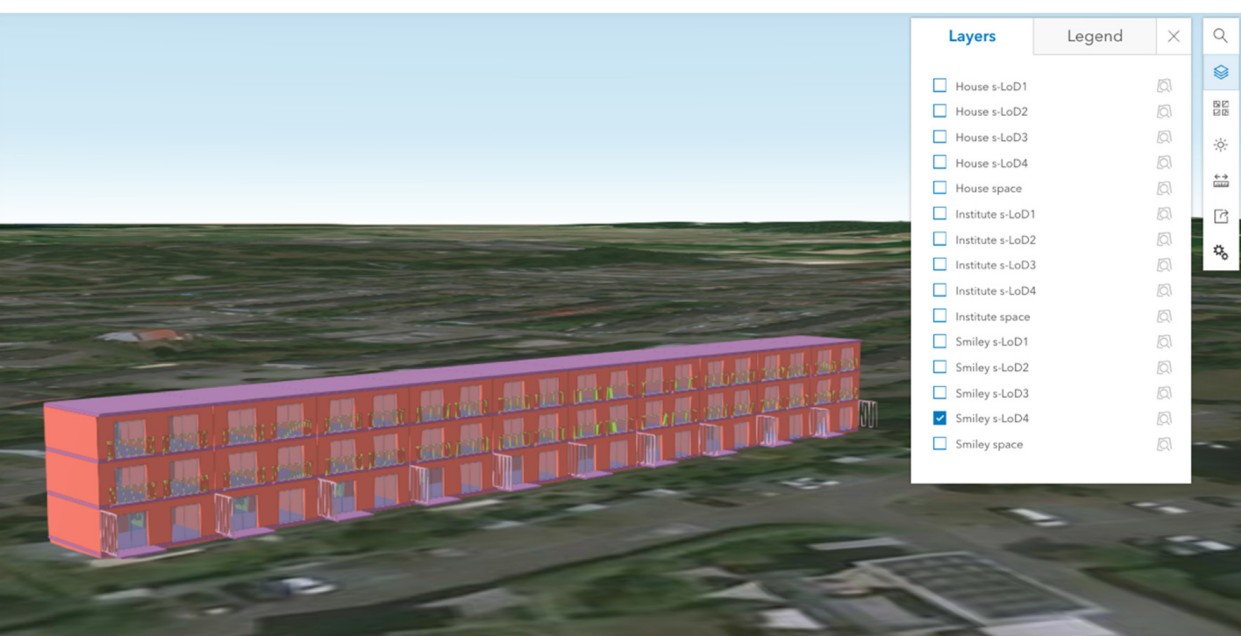

**Figure 14.** Smiley building of s-LoD4 on ArcGIS Online.

### 4.2. Effectiveness of Model Simplification

#### 4.2.1. Whole Building Model

The file sizes of these models are presented in Table 7. For each building, the absolute file size is given in Kilobyte (KB). It can be observed that for a specific building, the implementation of s-LoD can significantly reduce the file size. In terms of RRS1, from s-LoD4 to s-LoD1, the file size of all the three building models was reduced by 99.9%. In terms of RRS2, from s-LoD4 to s-LoD3, the file size was reduced by a percentage between 69.3% and 83.4%, and for s-LoD3 to s-LoD2, the reduced file size is between 87.1% and 98.3%.

**Table 7.** Size of models from s-LoD1 to s-LoD4 for house, institute and Smiley building.

| Model | Size | s-LoD4 | s-LoD3 | s-LoD2 | s-LoD1 |
|---|---|---|---|---|---|
| House | Absolute size | 1588.4 KB | 264.4 KB | 20.6 KB | 1.9 KB |
| | RRS1 | 100.0% | 83.4% | 98.7% | 99.9% |
| | RRS2 | NA | 83.4% | 92.2% | 90.8% |
| Institute | Absolute size | 3818.9 KB | 1064.9 KB | 137.1 KB | 2.8 KB |
| | RRS1 | 100.0% | 72.1% | 96.4% | 99.9% |
| | RRS2 | NA | 72.1% | 87.1% | 98.0% |
| Smiley | Absolute size | 16,146.5 KB | 4950.7 KB | 85.4 KB | 14.6 KB |
| | RRS1 | 100.0% | 69.3% | 99.5% | 99.9% |
| | RRS2 | NA | 69.3% | 98.3% | 82.9% |

#### 4.2.2. Windows, Doors, and Walls

Table 8 shows comparison between the simplified and non-simplified windows, doors and walls, using five instances. After simplification, windows and doors that originally contain multiple parts are generalized into one, and the file size can be reduced by up to 97.3% for doors, between 76.6% and 86.7% for windows, and around 47.0% for walls.

**Table 8.** Geometry simplification for over-detailed objects doors, windows, and walls.

| | Non-Simplified | | | Simplified | | | Reduced by |
|---|---|---|---|---|---|---|---|
| | Shape | Quantity of Parts | File Size | Shape | Quantity of Parts | File Size | |
| Door 1 |  | 32 | 70.4 KB |  | 1 | 1.9 KB | 97.3% |
| Door 2 |  | 14 | 70.1 KB |  | 1 | 1.9 KB | 97.2% |
| Window 1 |  | 7 | 14.3 KB |  | 1 | 1.9 KB | 86.7% |
| Window 2 |  | 11 | 127.8 KB |  | 1 | 29.9 KB | 76.6% |
| Wall |  | 4 | 16.8 KB |  | 1 | 8.9 KB | 47.0% |

A comparison in overall file size is also given in Table 9. It can be noticed that, after simplification, the overall file size of windows was reduced by a percentage between 80.6% and 82.9%, and between 67.8% and 97.7% for doors.

**Table 9.** File size of windows and doors before and after simplification.

| | Windows | | | Doors | | |
|---|---|---|---|---|---|---|
| | Original | Simplified | Reduced by | Original | Simplified | Reduced by |
| House | 349.7 KB | 68.0 KB | 80.6% | 132.8 KB | 5.4 KB | 95.9% |
| Institute | 1284.4 KB | 219.8 KB | 82.9% | 255.4 KB | 82.2 KB | 67.8% |
| Smiley | 498.8 KB | 85.4 KB | 82.9% | 7857.7 KB | 181.4 KB | 97.7% |

*4.3. Reliability of the Proposed Methods*

This study proposed to use semantic information when simplifying building models. However, despite our efforts to select quality models for method validation, problems regarding semantic information were encountered. The main problem is missing or erro-

neous semantic information. This issue has caused trouble for this study, especially when determining the externality of objects and obtaining the building story height, which is vital for generating s-LoD1, s-LoD2 and s-LoD3 models. This problem is, however, manageable during the model production stage, which has been explained in the discussion section.

### 4.3.1. Determining Externality of Objects

Two methods were proposed in this study for determining the externality of building elements, but it is found that, due to information loss, neither of them was problem-free. The first method, referred to as M1, uses the *IsExternal* attribute, whereas the second method, referred to as M2, uses the *IfcRelSpaceBoundary* relationship. Table 10 presents the results of the assessment of these two methods.

**Table 10.** Number of external, internal and undefined objects for each model identified by M1 and M2.

| Model | Class | External Objects | | Internal Objects | | Not Defined Objects | | Total | |
|---|---|---|---|---|---|---|---|---|---|
| | | M1 | M2 | M1 | M2 | M1 | M2 | M1 | M2 |
| House | IfcBeam | 0 | - | 3 | - | 1 | - | 4 | - |
| | IfcDoor | 0 | 2 | 0 | 3 | 5 | 0 | 5 | 5 |
| | IfcMember | 0 | - | 0 | - | 42 | - | 42 | - |
| | IfcRailing | 0 | - | 0 | - | 2 | - | 2 | - |
| | IfcSlab | 0 | 3 | 0 | 1 | 4 | 0 | 4 | 4 |
| | IfcStair | 0 | - | 1 | - | 0 | - | 1 | - |
| | IfcWallStandardCase | 0 | 8 | 0 | 5 | 13 | 0 | 13 | 13 |
| | IfcWindow | 0 | 11 | 0 | 0 | 11 | 0 | 11 | 11 |
| | Sub total | 0 | - | 4 | - | 78 | - | 82 | - |
| Institute | IfcColumn | 0 | - | 0 | - | 2 | - | 2 | - |
| | IfcDoor | 0 | 1 | 0 | 76 | 77 | 0 | 77 | 77 |
| | IfcRailing | 0 | - | 0 | - | 12 | - | 12 | - |
| | IfcSlab | 0 | 22 | 0 | 4 | 26 | 0 | 26 | 26 |
| | IfcStair | 0 | - | 0 | - | 4 | - | 4 | - |
| | IfcWallStandardCase | 0 | 44 | 0 | 77 | 121 | 0 | 121 | 121 |
| | IfcWindow | 0 | 206 | 0 | 0 | 206 | 0 | 206 | 206 |
| | Sub total | 0 | - | 0 | - | 448 | - | 448 | - |
| Smiley | IfcBeam | 10 | - | 0 | - | 0 | - | 10 | - |
| | IfcColumn | 20 | - | 0 | - | 0 | - | 20 | - |
| | IfcDoor | 70 | 77 | 90 | 93 | 10 | 0 | 170 | 170 |
| | IfcRailing | 118 | - | 0 | - | 2 | - | 120 | - |
| | IfcSlab | 90 | 90 | 25 | 30 | 5 | 0 | 120 | 120 |
| | IfcStair | 0 | - | 30 | - | 0 | - | 30 | - |
| | IfcWall * | 152 | 145 | 110 | 122 | 19 | 14 | 281 | 281 |
| | IfcWallStandardCase | 141 | 145 | 110 | 122 | 19 | 3 | 270 | 270 |
| | IfcWindow | 80 | 80 | 0 | 0 | 0 | 0 | 80 | 80 |
| | Sub total | 540 | - | 255 | - | 36 | - | 831 | - |

\* *IfcWallStandardCase* instances are included.

It can be noticed that the *IsExternal* attribute is not well managed in those models, especially for the institute building, where this attribute is not assigned at all, and all objects are categorized as undefined. In the house model, 78 of the 82 instances are undefined, and in the Smiley building, there are only 36 undefined objects out of 831. On the contrary, the *IfcRelSpaceBoundary* relationship is better managed than the *IsExternal* attribute, despite the fact that it can only be applied to boundary-related building elements. For the house model and the institute model, M2 managed to effectively identify external objects, but for the Smiley building, it failed to distinguish 14 wall instances.

### 4.3.2. Retrieving Building Story Heights

Building story height is important for the generation of the s-LoD1 model and the external walls of the s-LoD2 and s-LoD3 models. There are two ways to acquire building

story heights. The first is the sum of the heights of walls and floors on each story. The second is the *GrossHeight* attribute of building story, which is specified in the base quantity set (Qto_BuildingStoreyBaseQuantities). Table 11 presents the building story height of each IFC model using these two methods.

**Table 11.** Building story height acquired from two sources.

|        |                   | Story 1 | Story 2 | Story 3 | Story 4 | Story 5 |
|--------|-------------------|---------|---------|---------|---------|---------|
| House  | Floor height (m)  | **0.20** | **0.20** | -       | -       | -       |
|        | Wall height (m)   | **2.70** | **3.50** | -       | -       | -       |
|        | Gross height (m)  | **2.70** | **2.00** | -       | -       | -       |
| Institute | Floor height (m) | 0.30  | 0.30    | 0.30    | 0.30    | 0.30    |
|        | Wall height (m)   | 2.70    | 2.70    | 2.70    | 2.70    | 2.70    |
|        | Gross height (m)  | 3.00    | 3.00    | 3.00    | 3.00    | 3.00    |
| Smiley | Floor height (m)  | **0.25** | 0.18    | 0.18    | 0.18    | **0.20** |
|        | Wall height (m)   | **2.38** | 2.60    | 2.52    | 2.52    | **NA**  |
|        | Gross height (m)  | **2.56** | 2.78    | 2.70    | 2.70    | **2.70** |

It can be noted that there is inconsistency in height in the house model and Smiley building model, which has been highlighted. Take the Smiley building for example, for story 1, the sum of floor height and wall height is 2.63 m, while the value retrieved from the *GrossHeight* attribute is 2.56 m. In cases where there was inconsistency, the sum of floor heights and wall heights was used.

### 4.3.3. Influence of Erroneous Semantic Information

Missing or erroneous semantic information in IFC models has an adverse influence on this study. For example, for the house model, due to the incorrect story height of the second story, the initial s-LoD1 model (Figure 15a) was lower than it should have been. For the institute model, the floor and external slabs (Figure 15c) have to be manually separated, while these two types of slabs have been correctly differentiated in the Smiley model (Figure 15b).

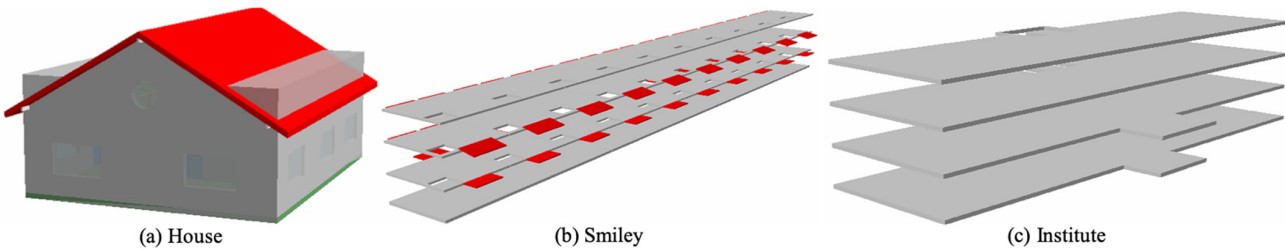

(a) House                          (b) Smiley                          (c) Institute

**Figure 15.** (**a**) Incorrect s-LoD1 model for house, (**b**) semantically differentiated slabs in Smiley building model, and (**c**) not semantically differentiated slabs in institute building model.

### 4.4. Assessing Solid Models by Comparing with Surface Models

To assess the solid model, a comparison was conducted with the CityGML model by using the house model in terms of appearance and file size. The house model in the CityGML format, from LoD1 to LoD4, was obtained from [56]. The LoD4 CityGML model was preprocessed using FME to ensure that same building elements are included.

#### 4.4.1. Model Appearance

Table 12 presents the (s-)LoD1 to (s-)LoD4 house models. CityGML models were visualized using FZK Viewer 5.1 and shapefile models were visualized in ArcScene 10.5. Some building components have been removed to show the interior structure. In (s-)LoD1, there is no difference in appearance except color. In (s-)LoD2 and (s-)LoD3, the difference between surface model and solid model can be obviously observed. The former uses

surfaces to represent boundary-related objects, while the latter uses solids, which are more intuitive.

**Table 12.** CityGML and shapefile models from (s-)LoD1 to (s-)LoD4.

| (s-)LoD | Exterior | | Interior | |
|---|---|---|---|---|
| | CityGML | Shapefile | CityGML | Shapefile |
| 1 |  | | | |
| 2 | | | | |
| 3 | | | | |
| 4 | | | | |

### 4.4.2. File Size

Table 13 shows the file size of shapefile and CityGML models in each (s-)LoD. In general, shapefile uses less storage space than CityGML in (s-)LoD1, (s-)LoD3 and (s-)LoD4. However, in (s-)LoD2, CityGML uses slightly less storage space. The file size of the CityGML model of the same house generated by Donkers et al. [6] was also obtained for comparison, which is 716.8KB, almost three times larger than this study.

**Table 13.** File size of the shapefile and CityGML model of house in each (s-)LoD.

| | (s-)LoD4 | (s-)LoD3 | (s-)LoD2 | (s-)LoD1 |
|---|---|---|---|---|
| Shapefile | 1605.9 KB | 262.8 KB | 22.2 KB | 1.9 KB |
| CityGML | 4099.5 KB | 283.4 KB | 16.1 KB | 7.4 KB |
| CityGML * | NA | 716.8 KB | NA | NA |

* from Donkers et al. [6].

## 5. Discussion

This study developed a framework to convert IFC models into solid models at various s-LoDs, which presents a new way to use BIM models in GIS (see Figure 16). One contribution of this study is taking model simplification into consideration, which was not considered by previous studies [18,43,44,57]. In addition, there are several new discoveries in this study regarding BIM/GIS integration and building model simplification.

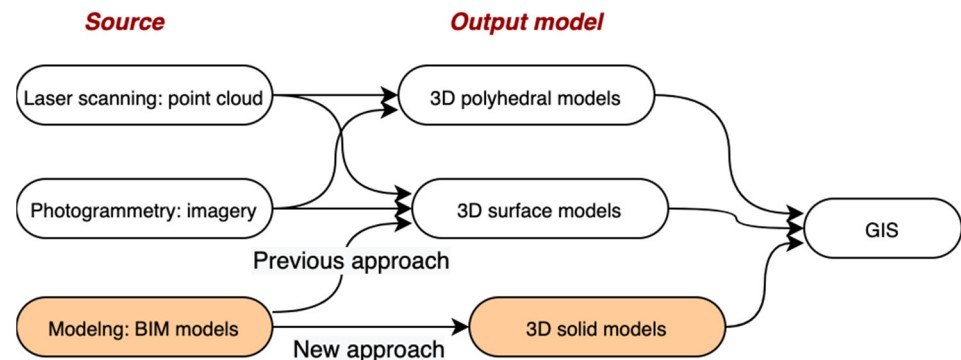

**Figure 16.** A new approach for using Building Information Modeling (BIM) models in Geographic Information System (GIS).

### 5.1. Adopting Solid Model to Facilitate BIM-to-GIS Data Conversion

The use of CityGML has mainly brought two challenges to BIM/GIS integration: the problematic solid-to-surface conversion and semantics transfer [6,16]. The adoption of the solid model can efficiently avoid these problems, as shown in Figure 17.

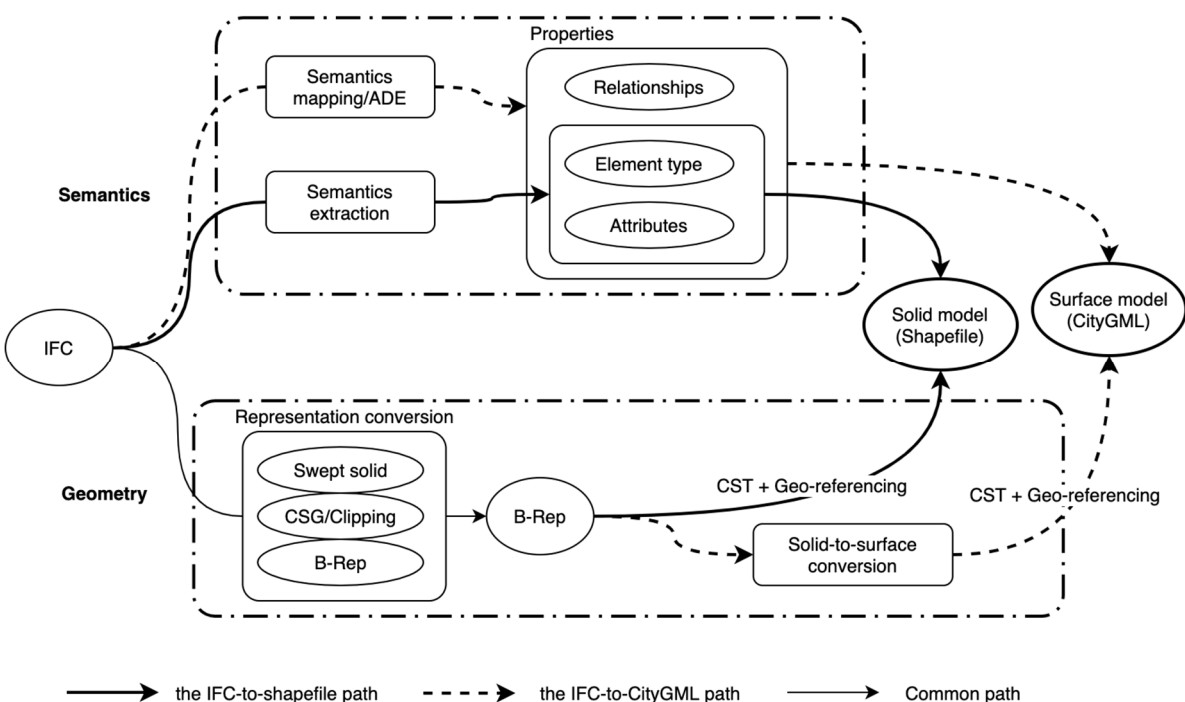

**Figure 17.** Paths for BIM-to-GIS data conversion.

The use of solid models also enables the IFC semantic information to be fully utilized in model simplification. For example, the attribute *IsExternal* can be used to identify exterior objects. For the IFC-to-CityGML conversion, due to the difference in the basic geometric unit (solid vs. surface) between IFC and CityGML, the semantic information in IFC was less helpful in CityGML model simplification.

### 5.2. New Opportunities and Challenges Brought by BIM to Model Simplification

Adopting BIM, or IFC, as source of 3D building models for the geospatial industry has brought new opportunities for building model simplification. The difference between the traditional approach and the proposed (new) approach in model simplification has been summarized in Table 14.

**Table 14.** Building model simplification in geospatial industry and in BIM/GIS integration.

| | Geospatial Industry (Traditional Surface-Based) | BIM/GIS Integration Approach | |
| --- | --- | --- | --- |
| | | Traditional Approach (Surface-Based) | Proposed Approach (Solid-Based) |
| Source model | High-LoD surface models | Solid models from BIM | Solid models from BIM |
| Target model | Surface models | Surface models | Solid models |
| Conversion type | Surface-to-surface | Solid-to-surface | Solid-to-solid |
| Geometry conversion pattern | High-to-low LoD conversion [22] | 1. High-to-low LoD conversion [16] 2. Generation from parameters [6] | Generation from parameters [43,44,57] |
| Information used | Geometric information | Geometric information and part of semantic information | Geometric information and full semantic information |

In the geospatial industry, the traditional simplification approach is a high-detail-to-low-detail conversion. Building simplification is realized by eliminating points from existing explicit models [19–21]. In this approach, the geometric information is the main information available, which makes this conversion full of challenges and prone to errors [6,16]. Additionally, the shape of building would affect the simplification procedure, which means that different algorithms are needed to deal with different building types, as shown in the study by Kim and Li [24]. BIM has brought a new approach for building model simplification, this new simplification approach, instead of eliminating points, is about generating fewer points. In addition, the simplification process is less affected by the shape of building, due to the IFC standard that specifies the geometry of building elements.

The traditional high-to-low simplification approach can also be applied to BIM models, as presented in Figure 18, where a wall with a hole is used as illustration. To create a s-LoD2 wall from the initial wall using the traditional approach, a geometry engine, such as Open CASCADE Technology (OCCT), can be used to generate the final explicit wall via two steps first (i.e., sweeping and differencing), and then the method can be applied to close the hole. However, it is less efficient compared with the proposed new approach, where the s-LoD2 wall can be directly generated by sweeping without applying the differencing operation.

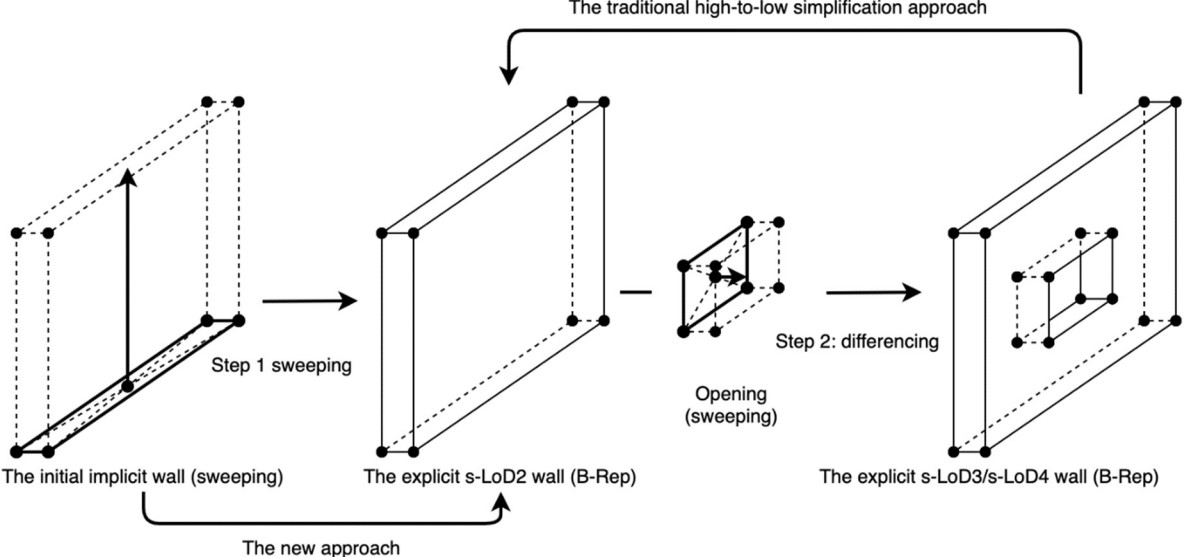

**Figure 18.** Wall simplification via the BIM/GIS integration approach and the traditional approach.

*5.3. The "Quality Issue" of BIM Models in BIM/GIS Integration*

The quality issue is not rare when BIM models are used in GIS, which has been encountered by many studies [6,10,58,59]. The root of this problem, from a perspective of

building information management, is that BIM models are project-specific. BIM models are intended to be produced to meet the certain needs of particular projects. According to ISO 19650, prior to the production process, the client (or the asset owner) must establish a clear project information requirement (PIR) to specify the need for information, such as the information content (geometry and semantics), production methods, and the date for information delivery. The client should only require information that is needed by the project, and any information not required by the project will be considered as waste. As a result, BIM models created for one project contain information specific to that project and may not suit the needs of other projects.

It appears that the BIM model production process, including the specification of information needed, has not been considered as part of the BIM/GIS integration. That is the cause of the "model quality issue" encountered by this study and many other studies. Please note that these models with a "quality issue" are probably problem-free for their original purpose.

In order to facilitate BIM/GIS integration, a set of information requirements should first be established to specify the need for information, and these requirements should be specified in the beginning of the project in the form of PIR. In addition, a dedicated model view definition (MVD) can be developed [60] for BIM/GIS integration. The concept of MVD is developed by buildingSMART [61], which is used to specify the certain entities/attributes to be included in specific IFC models.

### 5.4. Limitations and Future Work

This study is an early investigation into solid model simplification; the objective is not to develop a common method that would work in all situations, but to discuss a preliminary idea on how solid building models can be simplified. Therefore, the proposed method may not be applied to some building components, such as the revolving doors (see Figure 19) and curtain walls. Please also note that, according to the discussion on the "quality issue", it is obvious that not all existing BIM models contain the required semantic information, to which the proposed approach cannot be applied. In this regard, the proposed method is mostly future-oriented, in an attempt to make it more practical in the future to use BIM models in GIS.

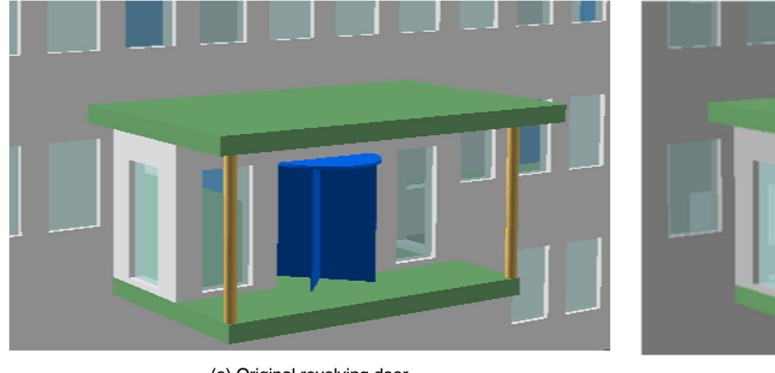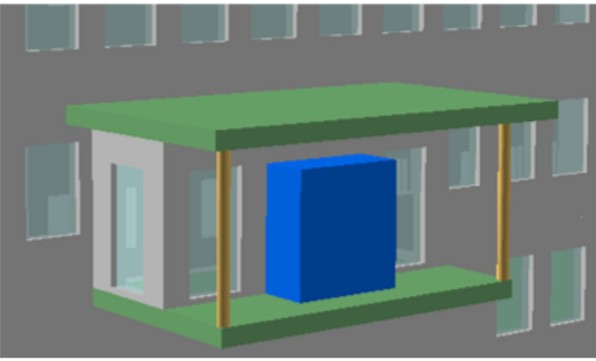

(a) Original revolving door          (b) Simplified revolving door

**Figure 19.** Simplification of the entrance door of the institute building model. (**a**) Original revolving door. (**b**) Simplified revolving door.

Another limitation is that this study mainly focused on the fundamental data conversion from BIM to GIS and the models generated in this study are currently primarily for visualization, which is the most fundamental requirement for BIM/GIS integration [59,62]. While some simple application scenarios can be provided, such as 4D simulation as presented in Zhu et al. [44], shadow analysis and dimension measuring that are built-in functions of ArcGIS Online, it is necessary to explore other application scenarios.

Other aspects that should be improved or investigated in the future are as follows. (a) Establishing information requirements for BIM/GIS integration and quality assessment for produced BIM models. BIM models are project-specific; it is thus necessary to take BIM/GIS data integration into consideration in the beginning of the project and establish a set of information requirements for that purpose. An automated method is also needed to ensure the quality of the created models. (b) Improving capability of the simplification methods by incorporating geometric information. When the required attributes are not available, 3D analysis (*Inside* 3D) that only uses geometric information can be applied to determine the externality of objects, which is more complex but would increase the capability of the proposed approach. (c) The definition of s-LoD references the LoD definition in CityGML, which is currently under revision and is probably to be modified to suit more application scenarios [63]. When this happens, the definition of s-LoD should be adjusted accordingly.

## 6. Conclusions

This study is one of the attempts trying to introduce BIM into the geospatial industry as a source of 3D building models. This study developed a framework for simplifying BIM models. The simplification was conducted at two levels, i.e., building level and building component level, during which the feasibility of using semantic information to facilitate the simplification process was investigated. At building level, a set of s-LoDs were specifically defined for solid building models. Unique challenges in applying such s-LoDs were identified and solved, including identifying external objects for s-LoD2 and s-LoD3, distinguishing various slabs, and generating valid external walls for s-LoD2 and s-LoD. At building component level, methods for simplifying over-detailed doors, windows and walls were proposed. Three BIM models were used to validate the proposed framework. The result shows that all three BIM models can be converted into s-LoD1 to s-LoD4 models, which justified the feasibility of the proposed methods.

This study has discovered or proved that: (1) It is feasible to use semantic information in building model simplification. The semantics-based simplification approach can be more efficient than the traditional high-to-low simplification approach. (2) Using the proposed simplification method, the building models can be simplified and converted into various s-LoDs, with file size being greatly reduced. In some cases, the reduction in file size of doors and windows can be up to 97.3%. (3) The BIM model production process should be considered as part of the BIM/GIS integration process. A set of information requirements specific to BIM/GIS integration should be developed, and these information requirements should be specified in the beginning of BIM projects. In addition, dedicated MVD can be developed to facilitate data exchange.

This study can contribute to GIS, especially 3D GIS, given that GIS needs building models in various LoDs to construct city models, while BIM can serve as an important source of building information, especially when more and more countries have mandated the use of BIM technology in public projects.

**Author Contributions:** Conceptualization, J.Z., C.A. and P.W.; methodology, J.Z.; software, J.Z.; validation, J.Z.; writing—original draft preparation, J.Z.; writing—review and editing, J.Z. and P.W.; funding acquisition, P.W. All authors have read and agreed to the published version of the manuscript.

**Funding:** This research was funded by the Australian Research Council, grant number DP180104026.

**Institutional Review Board Statement:** Not applicable.

**Informed Consent Statement:** Not applicable.

**Data Availability Statement:** Data available in a publicly accessible repository that does not issue DOIs. Publicly available datasets were analyzed in this study. This data can be found here: https://www.ifcwiki.org/index.php?title=IFC_Wiki (accessed on 27 October 2021).

**Acknowledgments:** The authors would like to thank the anonymous reviewers for their comments and suggestions that helped to improve the comprehensiveness and clarity of our paper.

**Conflicts of Interest:** The authors declare no conflict of interest.

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
