# Peer review of "A Semantics-Based Approach for Simplifying IFC Building Models to Facilitate the Use of BIM Models in GIS"

_remotesensing, doi:10.3390/rs13224727_

Round 1

Reviewer 1 Report

The authors describe their method to simplify IFC models for use in GIS. To be honest, I am very skeptical of this paper. The authors have applied some minor changes, but all the fundamental problems with it remain:

  1. Poor motivation, misrepresentation of the state of the art, and use of closed Shapefiles rather than an open format.
  2. Use of research IFC files from KIT that do not represent files from practice.

Reviewer 2 Report

A document with the comments is attached

Author Response

Thank you for your positive comments.

Reviewer 3 Report

This paper develops a framework to simplify BIM models of buildings in different levels of detail, to use building information in GIS environments. This is an interesting and active topic, not only for buildings but also for linear infrastructure, where GIS/BIM connection is even more relevant.

I believe this is a quite complete and consistent paper, well structured and with a clear representation of the information. I only have a few small comments that do not affect the core of this work’s content, so my recommendation is to accept this paper.

P2, L62: Here the aim of this work is highlighted, but it would be also convenient to explicitly add the contributions of this work in terms of novelity or uniqueness with respect other state-of-the-art works that are mentioned throughout this paper (and links to the contributions achieved as described in section 5).

I believe referring to both the step name and number in the headlines (e.g. 3.2 Building-level model simplification – Step 1) is not necessary. I would use only the name, as the step number can be retrieved from Figure 3 in any case.

It seems that here is a formatting error in Table 5, on the “IfcBeam” row, the word “institute” appears also as value.

Round 2

Reviewer 1 Report

I would like to state that I have no ill will towards the authors, whom I do not know.

However, as I have stated previously, I am very skeptical about this manuscript, which I believe has serious issues. I will explain my thought process on this.

  1. In terms of motivation, as the authors themselves state, the goal of converting BIM models to GIS models/formats is to use existing BIM models to fill data gaps (eg building interiors) or enhance existing models (eg more detail or more updated info).
  2. The authors, however, developed a methodology that requires geometries with attributes that are not set in practice, either by manually by employees of architecture/construction firms or automatically by software (please look at papers on BIM in practice about this).
  3. In order to show that the methodology "works", the authors must therefore rely on BIM models not generated from practice, and instead show their results using very old purpose-built models from KIT, which are very easy to convert using a variety of methods.

Because of the above, I believe that much more work needs to be done in order to show that the authors' methodology has been proven to work.

Author Response

This manuscript is a resubmission of an earlier submission. The following is a list of the peer review reports and author responses from that submission.

Round 1

Reviewer 1 Report

The authors propose a method to simplify IFC models for their use in GIS. The method almost entirely relies on semantics rather than on geometric simplification, which should lead to an efficient process with clean geometries and few vertices. At the same time, it suffers from the typical issues of semantics-based approaches, like errors due to bad/missing semantics.

In general, it is an okay paper. I think the authors have done novel work with a well-developed methodology and the manuscript documents it well enough. I was particularly impressed with the method used to close openings, which is very innovative and clever.

However, I think the paper has two serious outstanding issues:

  1. The authors are not adequately representing the state of the art in a few respects.

Most other work has focused on modelling surfaces because that is what is most practical for most applications. It is fine if the authors want to model solids instead, but claiming that this represents a new trend (backed only by a self-citation) or that “solid building models would have more potential than other types of building models in GIS applications” is misleading.

Similarly, characterising geometric simplification as something that was done before the advent of CityGML is not at all correct. Almost all researchers would agree that geometric simplification is part of the overall simplification process. Research continues on this topic.

Also, while some work has been done to convert IFC to Shapefiles, this is not really a desirable path for most applications. A Shapefile is a closed proprietary format with only rudimentary 3D support. Even in the closed Esri ecosystem, adequate 3D support is only obtained through i3s.

The Level of Development is very different to LoD in GIS. It is not related to geometric detail at all, but to the process that is usually followed to model a building.

Almost all IFC models are not really like LoD4. Much simpler IFC models are typically handed in for simpler buildings when required for permits. Also, most models usually don’t include many interior elements apart from the structural ones.

Despite what Table 1 says, IFC classes can use any representation types in practice. Also, the assertion that “sweeping is the most frequently used modeling method” needs to be backed by evidence.

  1. Regarding the methodology, I think that the authors are being extremely overoptimistic by relying only on the semantics used in IFC files. These are very, very frequently generated incorrectly by software. While the authors acknowledge this to an extent in Section 4, I do not believe this is given sufficient consideration, likely because the authors are relying on very simple BIM models generated by BIM experts from Karlsruhe. This sort of files do not resemble the ones used in practice at all.

To emphasise this point, in a recent discussion with a colleague (likely among the people who have dealt with the most BIM models in the world), he stated that he has never seen a real-world IFC file where the IsExternal attribute has been set correctly throughout. My own (lesser) experience is similar.

Based on the above, I think that the authors seriously need to test their methodology in other kinds of IFC files, preferably including some from practice. Without further tests, it is not really known whether the methodology can work with real files, which will have bad or missing semantics for some elements.

Reviewer 2 Report

Attend to the reviewer's instructions entered in the attached pdf document.
